# A Graphical Analysis Method of Guided Wave Modes in Rails

**Xining Xu** [1,2]**, Bo Xing** [1,]*⬦**, Lu Zhuang** [1]**, Hongmei Shi** [1,2] **and Liqiang Zhu** [1,2]⬦

[1]    School of Mechanical, Electronic, and Control Engineering, Beijing Jiaotong University, Beijing 100044, China; xnxu@bjtu.edu.cn (X.X.); 17121287@bjtu.edu.cn (L.Z.); hmshi@bjtu.edu.cn (H.S.); lqzhu@bjtu.edu.cn (L.Z.)

[2]    Key Laboratory of Vehicle Advanced Manufacturing, Measuring, and Control Technology, Beijing Jiaotong University, Ministry of Education, Beijing 100044, China

*    Correspondence: 15116333@bjtu.edu.cn

**Abstract:** The cross-section of a rail has a complex geometry, and there are many propagating modes of ultrasonic guided waves in a rail. The analysis of mode shapes or the cross-sectional wave structure is of high significance to the design of an appropriate wave excitation approach for long-range defect detection of a rail. Traditionally, the semi-analytical finite elements (SAFE) method is used to obtain ultrasonic guided waves' dispersion curves of a rail. Then, through solving the eigenvectors, it is able to calculate the displacement values of discrete nodes in three degrees of freedom (DOFs) and further obtain the wave structures. In this paper, a graphical analysis method of guided wave mode shapes is proposed. The displacements of each node in three DOFs are converted into Red Green Blue (RGB) image pixels, and the complex vibration vector data is expressed by an image. Therefore, the graphical analysis of mode shapes can be realized by using conventional image processing methods without the design of special data processing algorithms. This will improve the processing efficiency, and it is more intuitive and easier to analyze the vibration displacements represented by the image. The simulation results show that the proposed graphical analysis method can quickly and precisely locate the excitation position of the guided wave mode in the rail. By adopting image processing methods, such as the K-means clustering algorithm, the guided wave modes at a 35 kHz frequency in a rail are classified according to their mode shapes. Classification is essential for exploring the relations and fundamentals of vibrations in modes. The graphical analysis method proposed in this paper provides a novel method for the mode analysis of guided waves in rails.

**Keywords:** guided wave; mode; SAFE; mode of vibration; graphical; image processing

## 1. Introduction

With the continuous development of ultrasonic guided wave research, great progress has been made in the basic theory, numerical calculation, and simulation experiments. The application of ultrasonic guided wave technology is more and more extensive in the field of nondestructive testing. Cawley et al. [1,2] and Rose et al. [3,4] led their research groups to develop a pipeline detection system for non-destructive detection of buried pipelines based on ultrasonic guided wave technology. Loveday et al. [5,6] developed a rail detection system. Researchers have carried out non-destructive testing of waveguide mediums, such as flat plates [7,8], composite materials, pipes, rods [9], and rails [10,11], by using ultrasonic guided wave technology. The results obtained have greatly promoted the development and application of ultrasonic guided wave technology in petrochemical, aerospace, high-speed railway, and other industries.

Ultrasonic guided waves can cover the cross-section of the waveguide medium during propagation and have a higher detection efficiency [12]. However, guided waves have the characteristics of

multi-mode and dispersion, which also makes it complicated to use guided wave technology for non-destructive testing. The dispersion curves [13] describe the guided wave modes that can propagate at a certain frequency in a waveguide medium, and contain information, such as phase velocity, frequency, wavenumber, group velocity, and so on. It is important to develop a theoretical framework for analyzing and designing guided wave detecting equipment. There are many methods for solving the dispersion curves of waveguide media, and the semi-analytical finite elements (SAFE) method [14,15] is often used to solve the dispersion curves of guided waves in an arbitrary cross-section waveguide medium. After the finite element discretization on the cross-section of the waveguide medium, the wave process is expressed by the simple harmonic motion along the direction of the wave propagation and the wave equation is established based on the Hamiltonian principle of the variational method. The relationship between the wavenumber and frequency is obtained by solving the eigenvalue problems, and then the dispersion curves can be plotted.

When applying the SAFE method to solve the dispersion curves, the eigenvector is the displacement of each discrete node in the cross-section, and contains the shape of the guided mode. The eigenvector is used to analyze the mode shapes, which is important to study the excitation of the guided wave modes and develop a guided wave transducer. Takahiro Hayashi et al. [16] analyzed the process of Lamb wave propagation in metallic and resin plates loaded with water on a single surface. The structure of the Lamb wave was analyzed by the SAFE method. Takahiro Hayashi et al. [17] solved the dispersion curves of the guided waves in a rod, analyzed the mode shapes, and plotted the wave structures for a square bar. Alessandro Marzani [18] solved the dispersion equation of guided waves in cylinders and plotted a mode diagram of the n-order modes of the guided waves in a pipe. Ivan Bartoli et al. [19] plotted the mode shapes in a rail at low frequency, and analyzed the vibration characteristics of each mode based on the mode diagram.

In recent years, high-speed railways have developed rapidly [20]. Rails are important infrastructure components of railways. In order to guarantee the safe operation of high-speed railways, it is necessary to detect rails' internal defects and prevent potential fracture in advance.

The cross-section of the rail is complex. In the discrete cross-section, there are more nodes and elements, and the modes of vibration are complicated [21]. When analyzing the guided wave mode shapes in the rail, the method of plotting the mode displacement vector diagram is generally adopted [22]. The displacements of all nodes in the x, y, and z directions are listed in the mode vector matrix. In order to determine the energy distribution and maximum vibration position of each mode shape, it is necessary to design specific algorithms for the corresponding data analysis. This paper develops a graphical analysis method to analyze the vibration mode in the rail, which converts the node displacements into RGB pixel values, and expresses the mode vector diagrams by RGB color images intuitively. In this way, the existing image processing algorithms [23] can be used, which greatly enriches the analysis method for the mode of vibration. Converting the data processing of mode shapes into image processing simplifies the data analysis process and makes the mode of the vibration analysis more intuitive and simpler.

## 2. SAFE Method for Solving Vibration Mode Data of a Rail

### 2.1. SAFE Method

The SAFE method is used to achieve the mode shape information of guided wave modes in waveguide media by establishing the wave equation and solving the eigenvectors. The method only needs finite element discretization on the cross-section of the waveguide medium, while the displacements along the wave propagation direction are expressed by harmonic motion. In this paper, we consider a CHN60 rail, which is a 60 kg/m China rail. In Figure 1, the cross-section of the rail is defined as the *y-z* plane, and the wave propagation direction is the *x* direction.

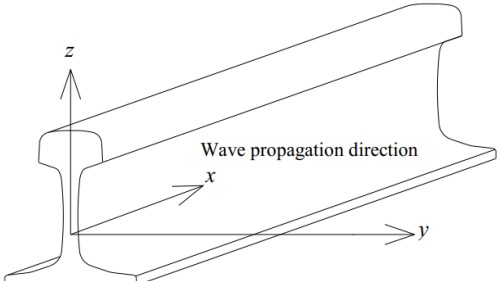

**Figure 1.** SAFE model of wave propagation.

Assuming that the displacement field of the guided wave propagating in the $x$ direction is a harmonic motion, the displacement, $u$, of any point in the waveguide medium is expressed as Equation (1) by a spatial function [24]:

$$\boldsymbol{u}(x,y,z,t) = \begin{bmatrix} u_x(x,y,z,t) \\ u_y(x,y,z,t) \\ u_z(x,y,z,t) \end{bmatrix} = \begin{bmatrix} U_x(y,z) \\ U_y(y,z) \\ U_z(y,z) \end{bmatrix} e^{i(\xi x - \omega t)} \tag{1}$$

where $\xi$ is the wavenumber and $\omega$ is the frequency.

Using the Partial Differential Equation (PDE) tool of MATLAB (version R2009a, the MathWorks, Inc., Natick, MA, USA), the cross section of the CHN60 rail is discretized by the triangular element. There are 177 nodes and 255 elements in the cross-section of the CHN60 rail as shown in Figure 2. Each element has three nodes and each node has three degrees of freedom, which correspond to displacements in the *x*, *y*, and *z* directions, respectively.

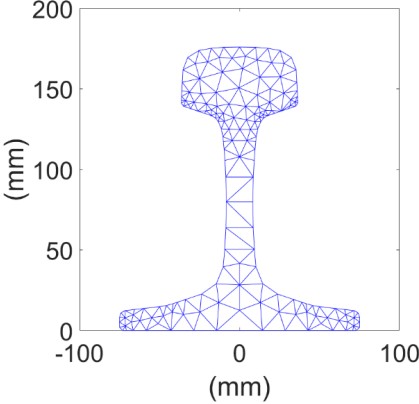

**Figure 2.** Discretization of a rail with triangle elements.

After the finite element discretization on the cross-section of the waveguide medium, the displacement of any point in the element is expressed as a form of the shape function [24] as Equation (2):

$$u^{(e)}(x,y,z,t) = \begin{bmatrix} \sum\limits_{k=1}^{n} N_k(y,z) U_{xk} \\ \sum\limits_{k=1}^{n} N_k(y,z) U_{yk} \\ \sum\limits_{k=1}^{n} N_k(y,z) U_{zk} \end{bmatrix}^{(e)} e^{i(\xi x - \omega t)} = \boldsymbol{N}(y,z)\boldsymbol{q}^{(e)} e^{i(\xi x - \omega t)} \tag{2}$$

where $U_{xk}$, $U_{yk}$, and $U_{zk}$ are the displacements of the nodes, $\boldsymbol{N}(y,z)$ is the shape function matrix, $\boldsymbol{q}^{(e)}$ is the node displacement vector, and $n = 3$ is the number of nodes in a triangular element. In Equation (2), the shape function is defined as:

$$N(y,z) = \begin{bmatrix} N_1 & & & N_2 & & & \cdots & & N_n & & \\ & N_1 & & & N_2 & & & \cdots & & N_n & \\ & & N_1 & & & N_2 & & & \cdots & & N_n \end{bmatrix} \tag{3}$$

$$q^{(e)} = \begin{bmatrix} U_{x1} & U_{y1} & U_{z1} & U_{x2} & U_{y2} & U_{z2} & \cdots & U_{xn} & U_{yn} & U_{zn} \end{bmatrix}^T \tag{4}$$

Based on the Hamiltonian principle, the general homogenization wave equation of ultrasonic guided waves in the CHN60 rail is derived [24] as Equation (5):

$$\left[ K_1 + i\xi K_2 + \xi^2 K_3 - \omega^2 M \right]_M U = 0 \tag{5}$$

where $K_1$, $K_2$, and $K_3$ are the stiffness matrix; and $M$ and $U$ are the mass matrix and nodal displacement vector, respectively

According to Equation (5), for a given wavenumber value, $\xi$, the value of the frequency, $\omega$, is calculated by calculating the eigenvalues. Thus, the relationship between the wavenumber and the frequency is determined. Subsequently, the phase velocity dispersion curves of the ultrasonic guided waves in teh CHN60 rail are depicted in Figure 3.

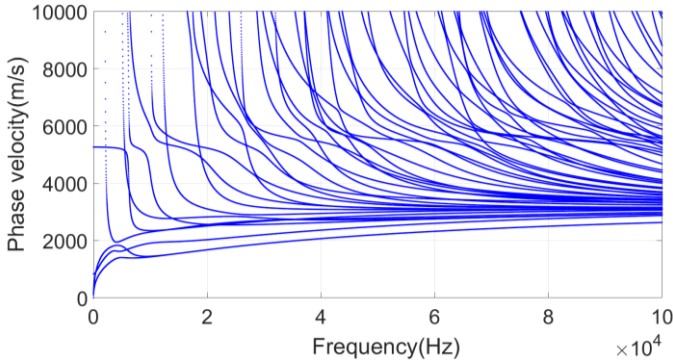

**Figure 3.** Phase velocity dispersion curves of the rail.

## 2.2. Solving the Vibration Mode Information

The material properties of the CHN60 rail are as follows: Density is 7800 kg/m³; modulus of elasticity is 210 GPa; Poisson's ratio is 0.3. The eigenvalues, frequencies, and eigenvectors are calculated by solving Equation (5). The eigenvalues reflect the dispersion characteristics of ultrasound guided waves and the eigenvectors represent the displacement of all nodes in the cross-section. At the frequency of 200 Hz, there are four ultrasonic guided modes in the CHN60 rail, as shown in Figure 4.

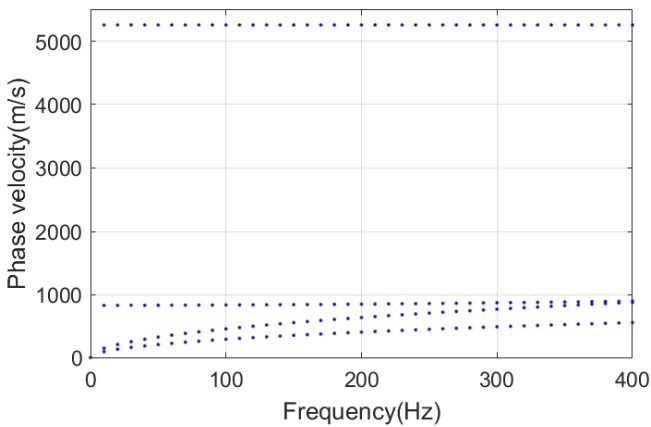

**Figure 4.** CHN60 rail's dispersion curves at low frequency.

The phase velocities of the four modes are 403 m/s, 634 m/s, 847 m/s, and 5263 m/s, respectively. Each mode shape is plotted according to the eigenvector, $\hat{U}$.

The eigenvalue, $\xi$, and the eigenvector, $\hat{U}$, are obtained by substituting $K_1, \hat{K}_2, K_3, \omega$, and $M$ into Equation (5) at the frequency, $f = 200$ Hz. The eigenvector, $\hat{U}$, represents the displacements of all nodes of the discrete rail cross-section. Figure 5a–d illustrates the mode shapes of the four modes, respectively. The colorized rail cross-section is the original size, and the red one is the mode shape.

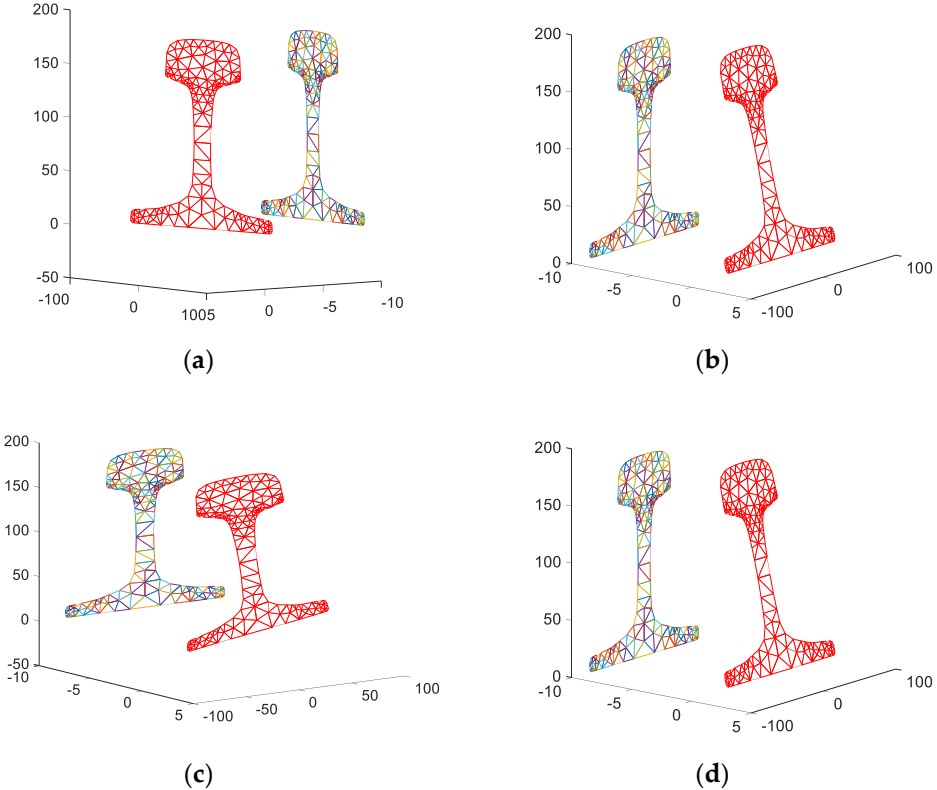

**Figure 5.** Mode shapes at 200 Hz. (**a**) The phase velocity is 403 m/s; (**b**) the phase velocity is 634 m/s; (**c**) the phase velocity is 847 m/s; (**d**) the phase velocity is 5263 m/s.

The vibration forms of each mode are able to be distinguished through an analysis of the mode shapes. Specifically, in the case of mode 1, there is no relative displacement inside the node, and the whole cross-section rotates along the vertical axis, thus the mode is the flexural horizontal mode; in mode 2, the cross-section is turning back and forth along the edge of the rail base, thus the mode is the flexural vertical mode. Mode 3 is the torsional mode, and mode 4 is the extensional mode. The three-dimensional vibration diagrams of all modes are shown in Figure 6.

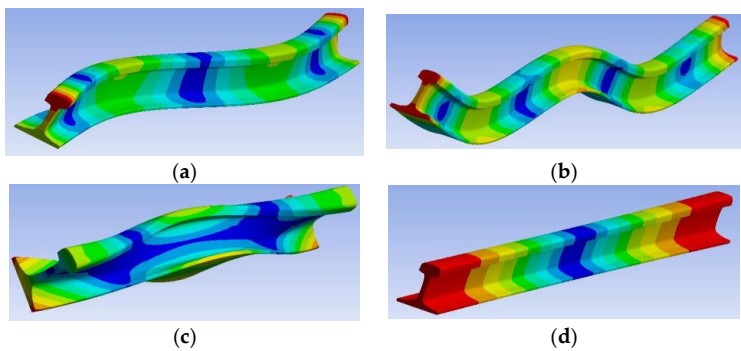

**Figure 6.** CHN60 rail's modes at low frequency. (**a**) Flexural horizontal mode; (**b**) flexural vertical mode; (**c**) torsional mode; (**d**) extensional mode.

The vibration characteristics of each mode can be known by analyzing the mode diagrams. It provides a data basis for the subsequent selection of guided wave modes suitable for long-distance propagation.

## 3. Graphical Analysis Method of the Mode Shape

At the low frequency, the number of guided wave modes in the rail is relatively small. With the increase of the frequency, the number increases gradually. Expressing the vector matrix data of the mode shapes in a graphical way can display the vibration characteristics of each mode more intuitively. Furthermore, processing is more efficient with a common image processing method.

Given a frequency, *f*, the eigenvector, $\boldsymbol{U}$, can be solved as Equation (5). It contains the displacement information of each node of all modes as Equation (6):

$$\boldsymbol{U}_n^m(x,y,z) = \begin{bmatrix} u_{1x}^1 & u_{1x}^2 & \cdots & u_{1x}^m \\ u_{1y}^1 & u_{1y}^2 & \cdots & u_{1y}^m \\ u_{1z}^1 & u_{1z}^2 & \cdots & u_{1z}^m \\ \vdots & \vdots & \cdots & \vdots \\ u_{nx}^1 & u_{nx}^2 & \cdots & u_{nx}^m \\ u_{ny}^1 & u_{ny}^2 & \cdots & u_{ny}^m \\ u_{nz}^1 & u_{nz}^2 & \cdots & u_{nz}^m \end{bmatrix} \tag{6}$$

The eigenvector, $\boldsymbol{U}_n^m(x,y,z)$, includes the mode shape information of m modes. There are *n* nodes, and each node has displacements in the *x*, *y*, and *z* directions in the discrete rail cross-section as shown in Figure 2. For a specific mode, the eigenvector solves the displacement values of *n* discrete nodes of all rail cross-sections. Taking the frequency of 35 kHz as an example, 20 guided wave modes can propagate in the rail, as shown in Table 1.

**Table 1.** Phase velocity and group velocity for each mode at the frequency of 35 kHz.

| Mode | Phase Velocity (m/s) | Group Velocity (m/s) |
| --- | --- | --- |
| 1 | 1984.05 | 2850.88 |
| 2 | 1983.80 | 2852.71 |
| 3 | 2286.06 | 3019.87 |
| 4 | 2592.53 | 2840.10 |
| 5 | 2725.71 | 2788.97 |
| 6 | 2743.72 | 2658.34 |
| 7 | 2737.57 | 3215.14 |
| 8 | 2919.28 | 3098.60 |
| 9 | 3104.71 | 2924.14 |
| 10 | 3297.15 | 2704.92 |
| 11 | 3389.87 | 2230.95 |
| 12 | 3708.61 | 2418.10 |
| 13 | 4194.78 | 1795.76 |
| 14 | 4167.81 | 2146.20 |
| 15 | 4955.74 | 2879.74 |
| 16 | 5554.76 | 4266.58 |
| 17 | 5862.23 | 2712.24 |
| 18 | 6796.88 | 2650.07 |
| 19 | 6415.38 | 2692.67 |
| 20 | 8155.23 | 3402.45 |

The shapes of mode 2, mode 3, mode 7, and mode 10 with phase velocities of 1983.80 m/s, 2286.06 m/s, 2737.57 m/s, and 3297.15 m/s, respectively, are plotted in Figure 7.

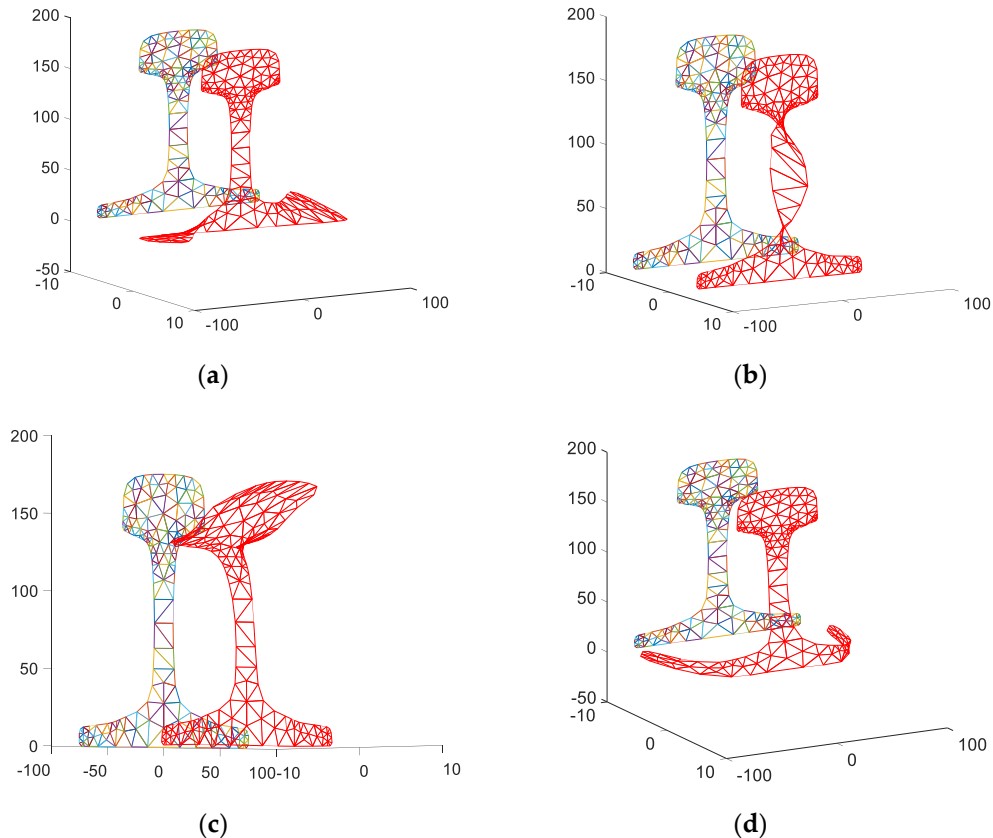

**Figure 7.** Guided wave mode diagrams at the frequency of 35 kHz. (**a**) The phase velocity of mode 2 is 1984.05 m/s; (**b**) the phase velocity of mode 3 is 2286.06 m/s; (**c**) the phase velocity of mode 7 is 2737.57 m/s; (**d**) the phase velocity of mode 10 is 3297.15 m/s.

According to the vibration mode information of the guided wave modes in the rail, the main vibration position and energy concentration point can be analyzed to determine the excitation position and direction of the mode. The orthogonal relationship between the modes can also be analyzed. However, these processing methods require related data processing on the mode matrix data without a fixed algorithm theory, and researchers need to design related algorithms according to the data characteristics.

At present, image processing technology has been widely used in various industries. There are many image processing methods, such as the histogram, binarization, gradient method, and so on. If the vibration mode vector data is converted into an image, the guided wave modes in the rail can be analyzed by the image processing method directly. The conversion idea is shown in Figure 8, which converts the displacement value of any point in the triangle element into RGB pixel values.

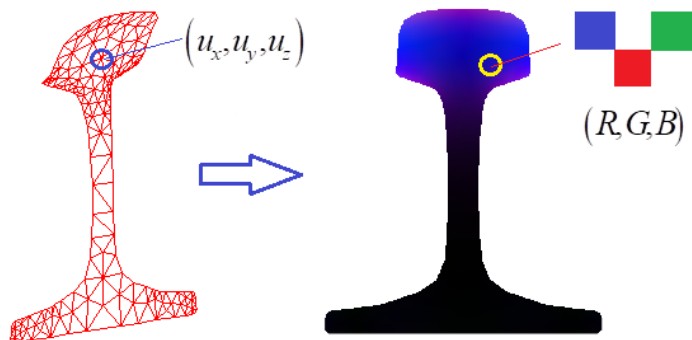

**Figure 8.** The conversion relationship between node displacements and RGB pixel values.

The conversion process includes the following steps:

(1) According to the rail size, the original image is generated, and the initial RGB pixel value is 0.
(2) Set the DPI value to generate the coordinate values of all the pixel points.
(3) Find the dot in the triangle elements of all coordinate values.
(4) Solve the displacement value of the current pixel point in the triangle elements by the difference equation.
(5) The displacement values of all the pixel points are converted into [0, 255] RGB pixel values by normalization.
(6) Plot the vibration mode image of the rail, RGB color, or gray image.

The width of the CHN60 rail base is 150 mm, and the height is 176 mm. The DPI value of the generated vibration image is set to 96. There are about 4 pixel points of 1 mm, and the image resolution is set to 601 × 705. Then, we created an image array I (601, 705, 3) and transformed 423,705 pixels in the image to the coordinate value ($y$, $z$) of the rail cross-section. Based on the coordinate values, the vibration displacement of each pixel is solved by the difference equation of a triangular element. First, we need to know whether the coordinate is located inside the triangle element. This is judged by the method of equal area, as shown in Figure 9.

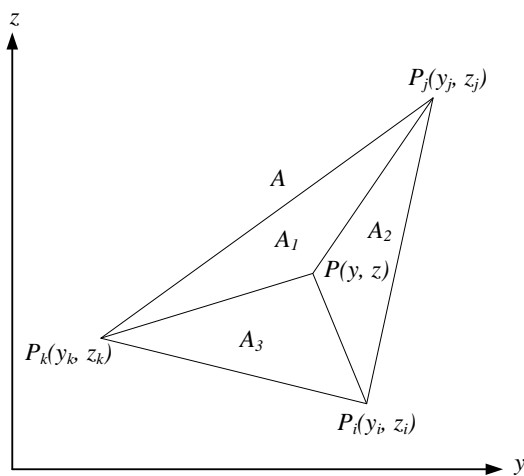

**Figure 9.** The determination method of the triangle element interior point.

The three vertices of the triangle element are $P_i$, $P_j$, and $P_k$, and its area is $A$. The points, $P$, $P_i$, $P_j$, and $P_k$, form three new triangles, with the areas of $A_1$, $A_2$, and $A_3$, respectively. If the point, $P$, is inside the triangle element, it should satisfy $A = A_1 + A_2 + A_3$. The coordinates of the three vertices of the triangle element are denoted as $P_i(y_i, z_i)$, $P_j(y_j, z_j)$, $P_k(y_k, z_k)$, and the area of the triangle is calculated using Equation (7):

$$A = \frac{1}{2}\left(y_i(z_j - z_k) + y_j(z_k - z_i) + y_k(z_i - z_j)\right) \tag{7}$$

For any triangular element, the displacement values of each node are solved from Equation (1). If the displacement values of the three nodes are known, the displacement value of any point inside the triangular element is obtained by the difference of Equation (2). For example, in Figure 9, the displacement values of three vertices (nodes) are, respectively: $(u_{ix}, u_{iy}, u_{iz})$, $(u_{jx}, u_{jy}, u_{jz})$, $(u_{kx}, u_{ky}, u_{kz})$. The displacement values of point $P$ in triangular elements $(u_x, u_y, u_z)$ are obtained as Equation (8):

$$\begin{cases} u_x = N_i u_{ix} + N_j u_{jx} + N_k u_{kx} \\ u_y = N_i u_{iy} + N_j u_{jy} + N_k u_{ky} \\ u_z = N_i u_{iz} + N_j u_{jz} + N_k u_{kz} \end{cases} \tag{8}$$

where $N_i$, $N_j$, and $N_k$ are triangular element shape functions as Equation (9):

$$\begin{cases} N_i = \frac{1}{2A}(\alpha_i + \beta_i y + \delta_i z) \\ N_j = \frac{1}{2A}(\alpha_j + \beta_j y + \delta_j z) \\ N_k = \frac{1}{2A}(\alpha_k + \beta_k y + \delta_k z) \end{cases} \tag{9}$$

where $A$ is the area of the triangle element, which is obtained as Equation (7), and $\alpha$, $\beta$, and $\delta$ are intermediate coefficients as Equation (10):

$$\begin{cases} \alpha_i = y_j z_k - y_k z_j \\ \beta_i = z_j - z_k \\ \delta_i = y_k - y_j \\ \alpha_j = y_k z_i - y_i z_k \\ \beta_j = z_k - z_i \\ \delta_j = y_i - y_k \\ \alpha_k = y_i z_j - y_j z_i \\ \beta_k = z_i - z_j \\ \delta_k = y_j - y_i \end{cases} \tag{10}$$

The displacement values of all pixel points are calculated according to Equation (8). They are normalized to obtain the pixel values of the interval [0, 255]. The images of the four mode shapes in Figure 7 are obtained by calculation, as shown in Figure 10.

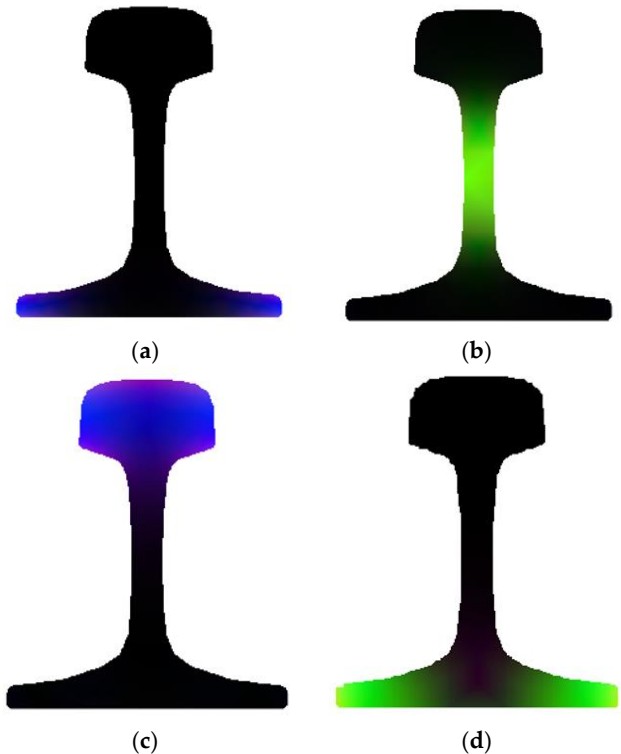

**Figure 10.** RGB image of four guided wave modes at the frequency of 35 kHz. (**a**) RGB image of mode 2; (**b**) RGB image of mode 3; (**c**) RGB image of mode 7; (**d**) RGB image of mode 10.

The main vibration position of the mode is clearly seen from Figure 10. When there is no vibration, the pixel value is 0, presented as the black image. According to the brightness, the displacement of the vibration is estimated. After generating the RGB image, the vibration mode can be analyzed by using the general image processing algorithm.

## 4. Image Processing Methods of the Mode Shape

After converting the vibration vector into the RGB image, the energy concentration position of the mode is determined by viewing the RGB image. As shown in Figure 10, the four modes at 35 kHz are selected. Figure 10a depicts mode 2 with a phase velocity of 1983.80 m/s, and its energy is concentrated at the base of the rail. Figure 10b shows mode 3 with a phase velocity of 2286.06 m/s, and its energy is concentrated at the rail web. Its maximum brightness is at the center of the rail web. Therefore, the crack at the rail web can be detected by this mode. The maximum vibrational positions of the other two modes are at the rail head and the rail base, respectively. Through simple image analysis and observation of the brightness distribution, it is possible to determine the location of the vibration energy concentration of each mode, and provide guidance for installation of the transducer when exciting specific modes. In the following, some simple image processing methods are adopted to analyze the mode image to obtain the vibration characteristics of the guided wave modes.

### 4.1. Histogram Processing

When plotting the vibration mode image of the rail, the displacement values in the $x$, $y$, and $z$ directions are converted into the pixel values of the RGB image, respectively. The vibration distribution of the guided wave modes in the $x$, $y$, and $z$ directions is analyzed with the histogram method. The histograms of three RGB colors are drawn, respectively. The abscissa represents the pixel value of the color from 0 to 255, and the ordinate represents the distribution of the pixel value. The more the distribution of the pixel values in the image is on the right side, the greater the vibration value of the mode in this direction.

It can be seen from Figure 11 that when the pixel value is bigger than 120, the blue pixels in the $z$ direction still has a 20~100 distribution in Figure 11d. Therefore, for mode 2, the histogram in the $z$ direction of the blue pixels distributes more in the region of the high pixel value, so the main direction of vibration is the $z$ direction.

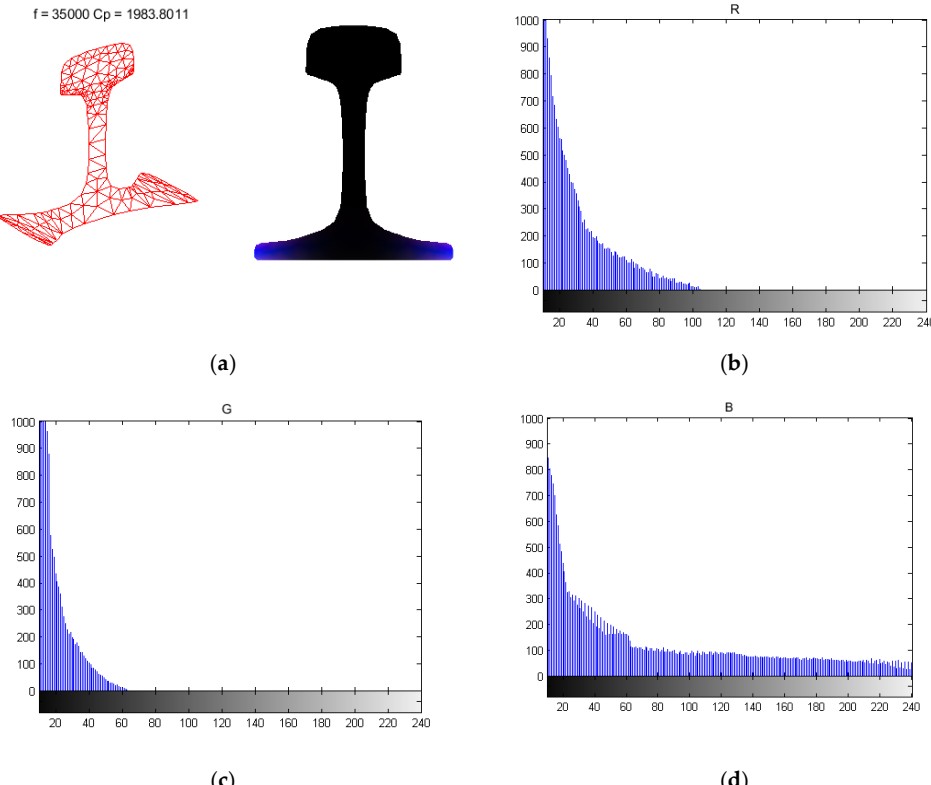

**Figure 11.** Histogram of mode 2 image. (**a**) RGB image of mode 2; (**b**) histogram of mode 2 in the $x$ direction; (**c**) histogram of mode 2 in the $y$ direction; (**d**) histogram of mode 2 in the $z$ direction.

In terms of mode 3, as the pixel value increases, only Figure 12c has a distribution in the high-pixel region. That is to say, the histogram along the *y* direction of the green pixels distributes more in the region of the high pixel value, so the main direction of vibration is the *y* direction.

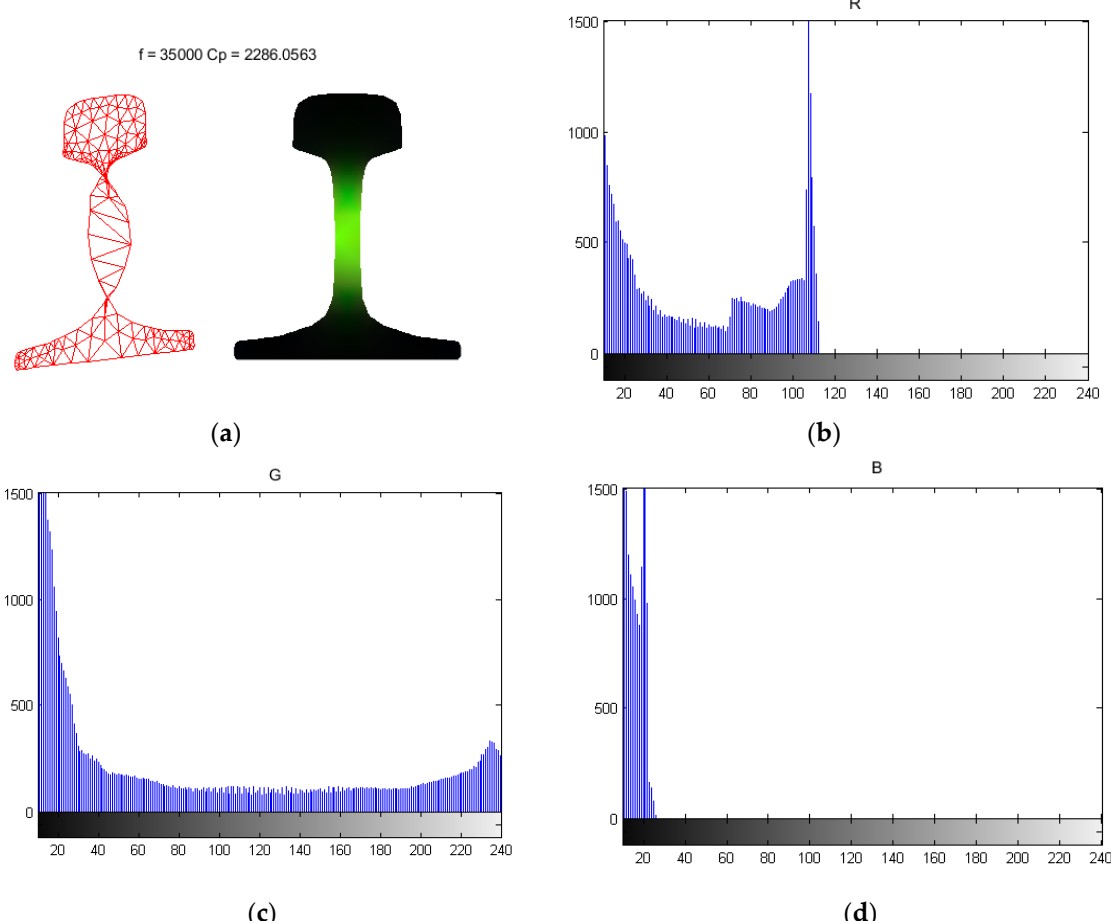

**Figure 12.** Histogram of mode 3 image. (**a**) RGB image of mode 3; (**b**) histogram of mode 3 in the *x* direction; (**c**) histogram of mode 3 in the *y* direction; (**d**) histogram of mode 3 in the *z* direction.

Similarly, in terms of mode 7, only Figure 13d has a distribution in the high-pixel range of 145 to 240. This is, the histogram of the blue pixels distributes more in the region of the high pixel value, so the main direction of vibration is the *z* direction.

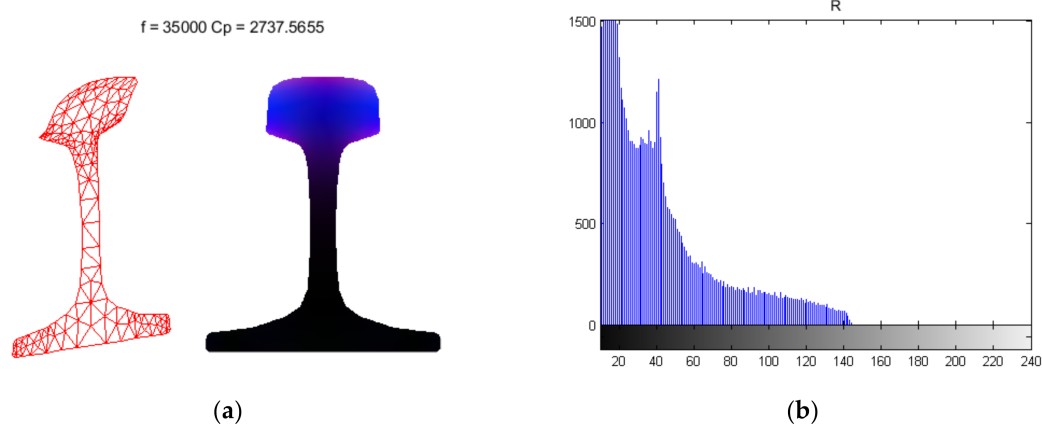

**Figure 13.** *Cont.*

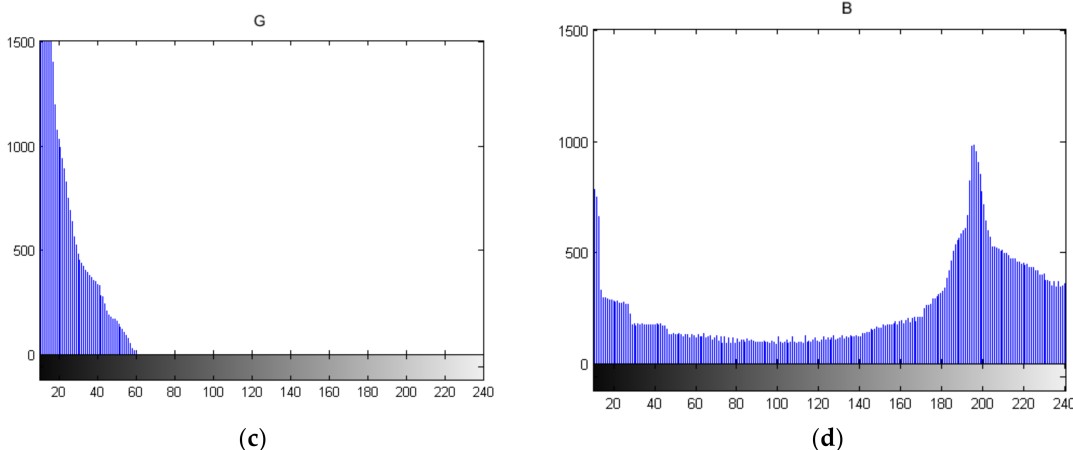

**Figure 13.** Histogram of mode 7 image. (**a**) RGB image of mode 7; (**b**) histogram of mode 7 in the *x* direction; (**c**) histogram of mode 7 in the *y* direction; (**d**) histogram of mode 7 in the *z* direction.

In terms of mode 10, when the pixel value is bigger than 180, the green pixels in the *y* direction still has the distribution in Figure 14c. That is to say, the histogram of the green pixels distributes more in the region of the high pixel value, so the main direction of vibration is the *y* direction.

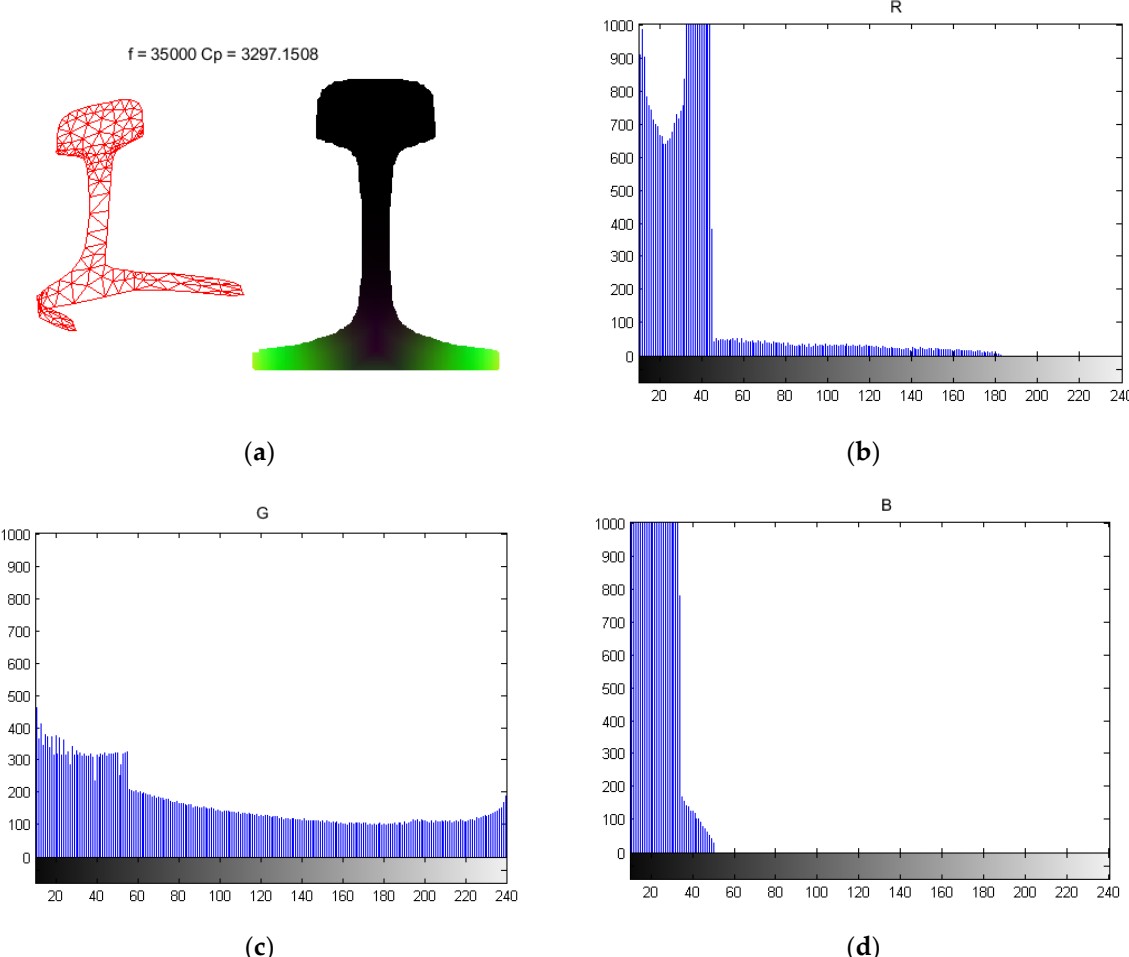

**Figure 14.** Histogram of mode 10 image. (**a**) RGB image of mode 10; (**b**) histogram of mode 10 in the *x* direction; (**c**) histogram of mode 10 in the *y* direction; (**d**) histogram of mode 10 in the *z* direction.

### 4.2. Image Gradient and Binarization

After obtaining the energy distribution area and the main vibration direction of a mode, it is necessary to estimate the maximum vibration area of this mode in order to determine the installation location of the transducer when exciting this mode. To deal with this issue, the binarization method of the image processing is used by setting a maximum pixel threshold. The pixel value is set to 1 if it is greater than the threshold and 0 if it is less than the threshold. The pixel value of the non-rail area is set to 0 in order to eliminate disturbances. The outline of the rail is plotted by the edge extraction method of the image gradient. The gradient edge extraction and binarization image of mode 3 are shown in Figure 15.

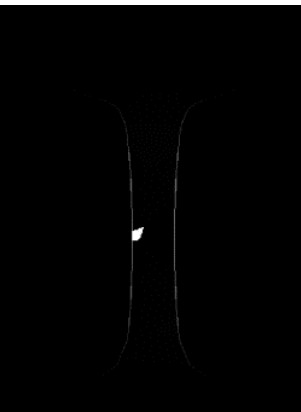

**Figure 15.** The gradient edge extraction and binarization image of mode 3.

After binarization, it can be seen that the maximum vibration point of mode 3 is in the rail web center. Therefore, in order to excite mode 3, a transducer is installed in the center of the rail web and excited in the $y$ direction.

In the same way, mode 7 is binarized. As shown in Figure 16, the maximum vibration point of mode 7 is located on both sides of the rail head in the $z$ direction.

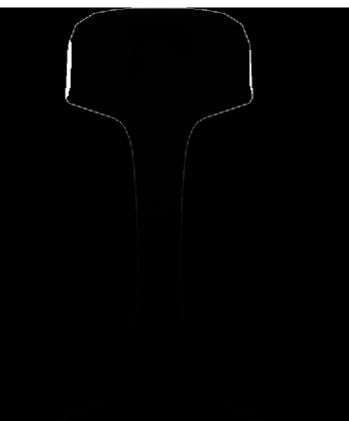

**Figure 16.** The gradient edge extraction and binarization image of mode 7.

### 4.3. Mode Classification Based on the K-Means Clustering Algorithm

In the study of guided waves, researchers usually classify guided wave modes according to the mode shapes. Lamb waves in plates are divided into symmetric and antisymmetric modes. The modes in the pipeline are divided into the longitudinal mode, torsional mode, and bending mode. Due to the complexity of the cross-section, the modes in the rail are divided into the longitudinal mode, torsional mode, and bending mode according to the mode shapes at low frequencies. However, the mode is very complicated at high frequencies, e.g., at 35 kHz, there are 20 modes in the rail as shown in Figure 17.

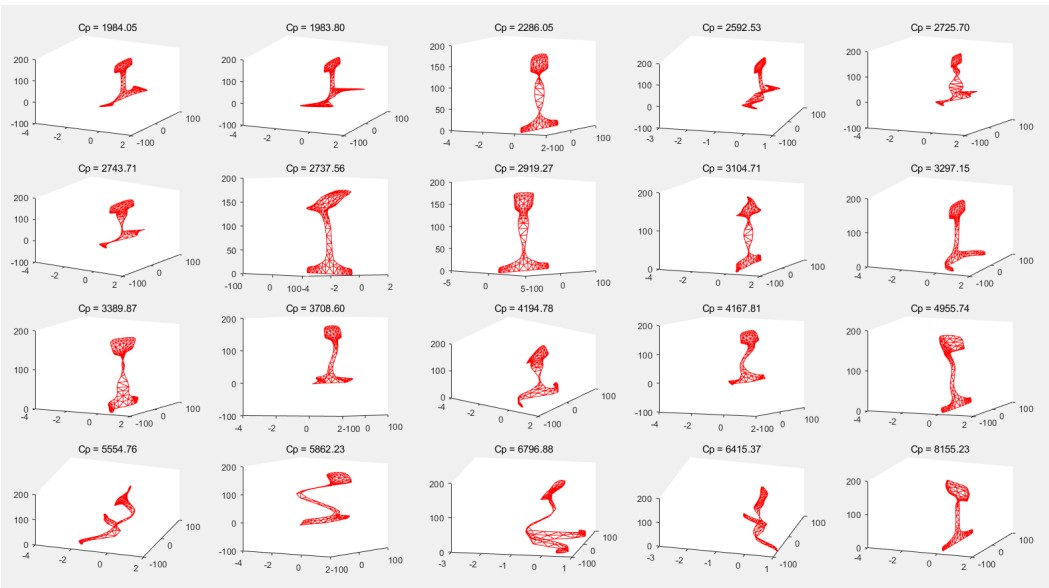

**Figure 17.** Guided wave mode shapes at the frequency of 35 kHz.

As clearly shown in the figure, the guided wave modes are complex and cannot be classified intuitively from the mode diagrams. Therefore, we converted the mode shapes in Figure 17 into RGB images as shown in Figure 18.

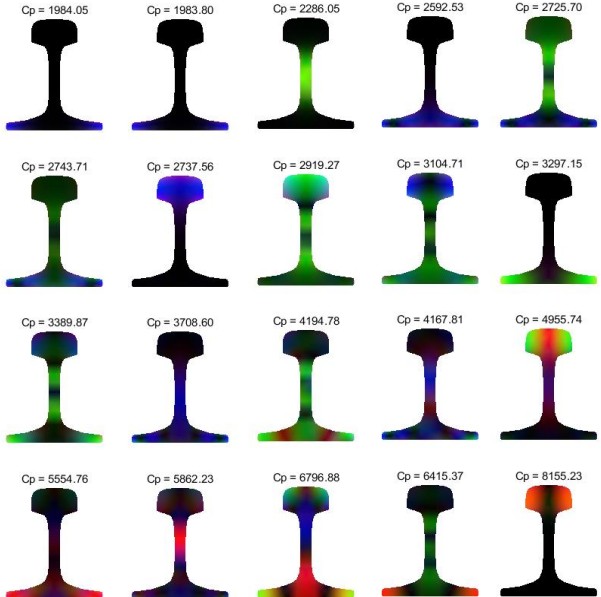

**Figure 18.** RGB images of guided wave modes at the frequency of 35 kHz.

According to the color image of Figure 18, the common image classification algorithms in image processing, such as the K-means clustering algorithm, can be used to classify the color image of modes. K-means is one of the clustering algorithms [25], where K represents the number of categories and means represents the mean. The K-means clustering algorithm divides similar data points by a preset K value and the initial centroid of each category. The optimal clustering results are obtained by iterative optimization of means.

Firstly, the RGB images of the guided wave modes in Figure 18 is compressed to reduce the dimension of image data processing. The original image resolution of the mode shape is $601 \times 705$, and the color image is compressed to $32 \times 32$ pixels as shown in Figure 19.

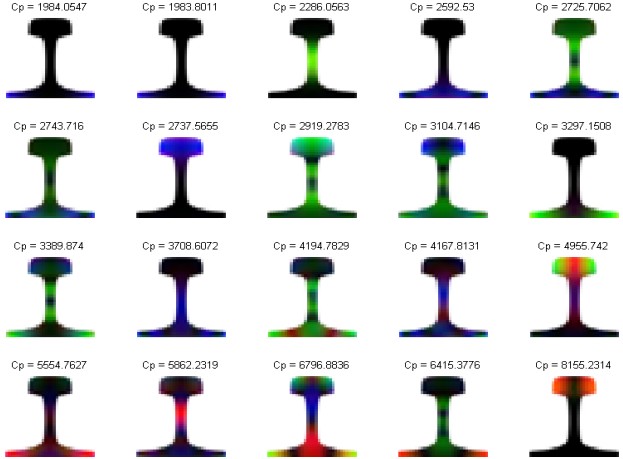

**Figure 19.** Mode compression image.

Each color image of the mode shape is expanded into a one-dimensional array. There are $32 \times 32$ pixels, each of which contains three RGB values. Then RGB images of 20 modes generate a $20 \times 3072$ raw data array shown as Equation (11):

$$
data = \begin{bmatrix}
R^1_{1,1} & G^1_{1,1} & B^1_{1,1} & R^1_{1,2} & G^1_{1,2} & B^1_{1,2} & \cdots & R^1_{32,32} & G^1_{32,32} & B^1_{32,32} \\
R^2_{1,1} & G^2_{1,1} & B^2_{1,1} & R^2_{1,2} & G^2_{1,2} & B^2_{1,2} & \cdots & R^2_{32,32} & G^2_{32,32} & B^2_{32,32} \\
\vdots & \vdots & \vdots & \vdots & \vdots & \vdots & \cdots & \vdots & \vdots & \vdots \\
R^{20}_{1,1} & G^{20}_{1,1} & B^{20}_{1,1} & R^{20}_{1,2} & G^{20}_{1,2} & B^{20}_{1,2} & \cdots & R^{20}_{32,32} & G^{20}_{32,32} & B^{20}_{32,32}
\end{bmatrix}
\tag{11}
$$

where $R^n_{x,y}, G^n_{x,y}, B^n_{x,y}, x, y = \{1, 2, \ldots 32, \}, n = \{1, 2, \ldots 20\}$ represent three color pixel values of the nth mode.

The array is classified by the K-means clustering algorithm with K = 5. Since the initial centroid of the K-means clustering algorithm is randomly selected, the results are inconsistent. Therefore, the classification results with the highest frequency are obtained through multiple classifications, as shown in Figure 20.

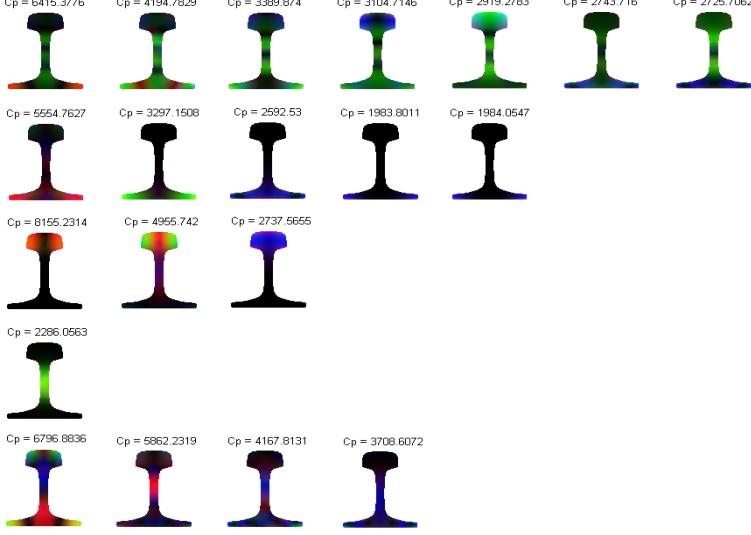

**Figure 20.** Mode diagrams' classification results.

In order to observe the classification results conveniently, the vibration displacement diagrams of each mode are plotted as shown in Figure 21.

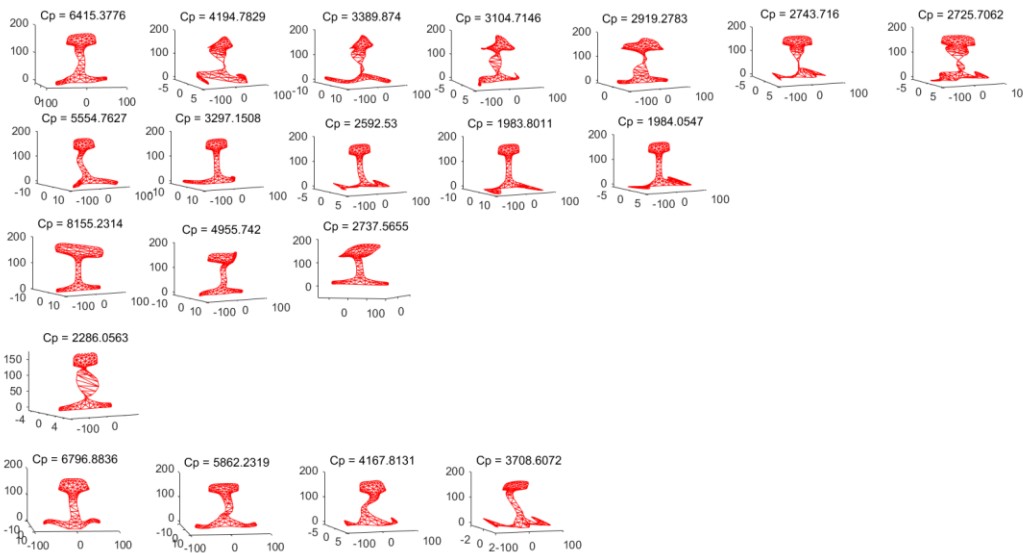

**Figure 21.** Mode displacement diagrams of the classification results.

It can be seen from Figure 21 that the K-means clustering algorithm divides the mode shapes into five categories at the frequency of 35 kHz in the rail. The first type of mode shape has large vibration over the whole rail cross-section. In the second type, the maximum vibration position is located at the rail base. The vibration is mainly located at the rail head in the third type. The fourth type is a torsional vibration. The vibration of the rail head and the rail base is small, and the vibration of the rail web is large. In the fifth type, the vibration position is in the rail web, and its vibration is S-type and non-torsional. By classifying the guided wave modes in rails at specific frequencies, the vibration relations and regularity of each mode are found. It is of certain theoretical guiding significance for the application of guided wave technology to detect internal defects of rails at different locations.

## 5. Mode Excitation Simulation

Taking mode 3 at frequency of 35 kHz as an example, it can be seen from Figures 10, 12, and 15 that the maximum vibration region of mode 3 is located at the rail web, the main vibration direction is the y direction, and the maximum vibration point is located at the center of the rail web. Therefore, as shown in Figure 22, the excitation point is applied in the *y* direction at the center of the rail web to verify whether mode 3 is excited.

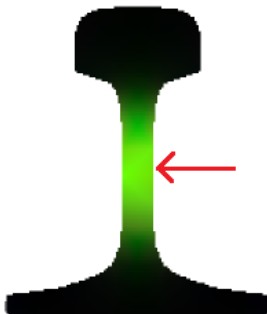

**Figure 22.** Mode 3 excitation simulation.

The CHN60 rail model with a length of 10 m was established by using the ANSYS (version 15.0, the ANSYS, Inc., Canonsburg, PA, USA) simulation software. In order to avoid the influence of the rail end echo on the simulation results, the excitation signal was applied to the rail periphery at 4 m from the left end of the rail model, and a data acquisition point was set between 0.8 m and 2.7 m from the excitation position with intervals of 5 mm. There were 380 synchronous acquisition points.

The phase velocity of the mode was calculated by the two-dimensional fast Fourier transform (2-D FFT). The simulation results were compared with the frequency-wavenumber dispersion curves solved by the SAFE method as shown in Figure 23.

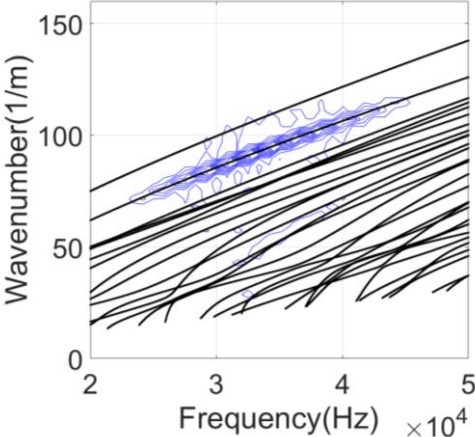

**Figure 23.** Excitation simulation results of mode 3.

We can see from Figure 23 that the contour map is the 2-D FFT results of mode 3 and the black lines are frequency wavenumber dispersion curves. After merging the two curves, the 2-D FFT results coincide well with the dispersion curve of mode 3. Thus, mode 3 is excited at this point in the $y$ direction. Therefore, the method of determining the direction and location of the mode excitation proposed in this paper can excite the expected mode.

## 6. Conclusions

Ultrasonic guided waves can propagate a long distance in the rail. They are usually used for on-line monitoring of rail internal defects, which provides early warning data for rail maintenance. There are many guided wave modes that can propagate in the rail, and their mode shapes are different. The analysis of the vibration regularity of guided wave modes is of great significance for mastering the propagation characteristics of guided waves and the efficient use of guided waves for non-destructive testing.

Based on the SAFE method, the dispersion curves of the guided waves in the rail were solved. The eigenvectors contain the vibration displacements of discrete nodes of the rail cross-section. Since there were no existing algorithms in the literature, it was necessary to design one for the analysis of the vibration mode data while using the traditional matrix data processing method. It requires numerous data computing and is not intuitive. Therefore, this paper proposed a processing method to convert the vibration mode data of the rail into the RGB image. Modes of vibration were visually displayed as an image. At the same time, it is realizable to directly use various existing algorithms of image processing. The mode shapes of the rail can be analyzed by image processing algorithms, such as the histogram, binarization, gradient solution, and image classification. By applying the proposed image processing method in the analysis of vibration mode data, the guided wave modes at a specific frequency were preliminarily classified. The energy concentration position of each mode was further determined. The simulation results proved that this method provides a solid theoretical reference for the design and installation of excitation transducers. When detecting rail cracks based on ultrasonic guided wave technology, this method can also guide the selection of a specific mode to detect a crack in different cross-section positions.

**Author Contributions:** Conceptualization, X.X.; methodology, X.X. and B.X.; validation, X.X. and B.X.; formal analysis, B.X.; writing, L.Z. (Lu Zhuang); review and editing, H.S. and L.Z. (Liqiang Zhu).

**Funding:** This research was funded by the Fundamental Research Funds for the Central Universities (2019JBM045).

**Acknowledgments:** The authors would like to thank Dong Lijing for her help in proofreading this paper.

**Conflicts of Interest:** The authors declare no conflict of interest.

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
