# Peer review of "A Graphical Analysis Method of Guided Wave Modes in Rails"

_applsci, doi:10.3390/app9081529_

Round 1

Reviewer 1 Report

It is well described paper.

Some comments:

1- Why authors decided to use triangular shape elements in modeling? If it has a proven effect on accuracy, it must be discussed in details with citing appropriate references.

2- How authors decided about the size of elements? Please discuss it in details.

3- Any experimental data to validate/verify the numerical model?

4-Spelling out the acronyms in their first use.

Author Response

Dear Professor:

Thank you for your kind comments on our paper entitled “A Graphical Analysis Method of Guided Wave Modes in Rails” (applsci-467361). We have checked the manuscript and made a revision according to the comments. Revised portion are marked in Red.

Reviewer 2 Report

Please find enclosed my comments.

Author Response

(The authors gave the same response as above.)
